# Embedding-based Instance Segmentation in Microscopy

**Manan Lalit** [1,2]                                          LALIT@MPI-CBG.DE

**Pavel Tomancak** [1,2,3]                              TOMANCAK@MPI-CBG.DE

**Florian Jug** [1,2,4]                                    FLORIAN.JUG@FHT.ORG

[1] *Center for Systems Biology Dresden (CSBD)*

[2] *Max Planck Institute of Molecular Cell Biology and Genetics*

[3] *IT4Innovations, VŠB - Technical University of Ostrava, Ostrava-Poruba, Czech Republic*

[4] *Fondazione Human Technopole, Milano, Italy*

## Abstract

Automatic detection and segmentation of objects in 2D and 3D microscopy data is important for countless biomedical applications. In the natural image domain, spatial embedding-based instance segmentation methods are known to yield high-quality results, but their utility for segmenting microscopy data is currently little researched. Here we introduce EMBEDSEG, an embedding-based instance segmentation method which outperforms existing state-of-the-art baselines on 2D as well as 3D microscopy datasets. Additionally, we show that EMBEDSEG has a GPU memory footprint small enough to train even on laptop GPUs, making it accessible to virtually everyone. Finally, we introduce four new 3D microscopy datasets, which we make publicly available alongside ground truth training labels. Our open-source implementation is available at https://github.com/juglab/EmbedSeg.

**Keywords:** instance segmentation, microscopy, spatial embeddings, deep learning

## 1. Introduction and Background

Instance segmentation of structures in microscopy images is essential for multiple purposes. In recent years, many Deep Learning (DL) based approaches to microscopy image segmentation have been proposed (Moen et al., 2019; Caicedo et al., 2019b; Schubert et al., 2018). Such methods can be divided into *Top-down* and *bottom-up* methods. Mask R-CNN (He et al., 2017), for example, is arguably the most prominent top-down method, designed to detect object instances via bounding-boxes. An additional refinement step produces a pixel-mask from multiple predicted bounding-box detections. Bottom-up methods, in contrast, are designed such that each pixel makes a prediction of either the object class it belongs to (Ronneberger et al., 2015), and/or the shape of the object instance it is part of (Schmidt et al., 2018; Neven et al., 2019; Hirsch et al., 2020). In a second phase, all methods need to consolidate their detections/predictions in order to obtain the final set of object instances. Mask R-CNN (He et al., 2017) or StarDist (Schmidt et al., 2018), for example, avoid multiple detections of the same object by employing non-maximum suppression on an instance associated confidence score. While DL-based methods helped to improve microscopy image data segmentation considerably, automated results are still subject to many errors that need to be addressed with manual post-processing.

An additional complication comes from differences between the domain of natural and microscopic images. While objects in natural images are typically either vertically or horizontally aligned, objects in microscopy typically have complex and unique shapes and are

randomly oriented. Hence, methods that employ axis-aligned bounding boxes, such as Mask R-CNN, tends to perform rather poorly. StarDist improves this shortcoming by assuming star-convexity of objects to be segmented. While being the key to success for some datasets, this assumption backfires when morphologically more complex shapes need to be segmented.

Another shortcoming of today's segmentation landscape is that most methods only operate on 2D image data. Methods to segment volumetric data (3D image data), despite desperately needed, are much less common. Existing 3D implementations either perform volumetric data segmentation by combining results on 2D slices (Stringer et al., 2020), or, if directly operating on 3D images, tend to require large and expensive GPU hardware (see *e.g.* Table 2).

Here we present EMBEDSEG[1], a variation of the inspiring work in (Neven et al., 2019), a very compact model for end-to-end instance segmentation. Each pixel predicts its own *spatial embedding*, *i.e.* another unique pixel location that is meant to represent the object this pixel is part of. Additionally, the network learns an instance-specific clustering band-width, later used to cluster embedding pixels into object instances. The segmentation mask of an object is defined by all pixels that point to the same cluster of embedding pixels. An additional *seediness score* for each pixel is predicted, indicating how likely it is for the respective pixel, and its associated clustering band-width, to represent an object instance.

We propose several modifications that greatly improve the performance of embedding-based instance segmentations on microscopy data: Importantly, EMBEDSEG is not limited to 2D images but can directly be trained and applied on volumetric 3D data. Instance segmentation results on three 2D and four 3D datasets are presented in Section 3 and Tables 1 and 2. Last but not least, we make all four used 3D datasets and their respective training labels publicly available[2].

## 2. Related Work and Proposed Method

Embedding-based segmentation methods have recently emerged in the context of multiperson pose estimation. (Newell et al., 2017) initially suggested a DL framework where each pixel predicts a *tag* or *embedding*. The proposed objective encourages pairs of tags to have similar values if and only if the corresponding pixels belonged to the same object. In the same year, (Brabandere et al., 2017) suggested a specific hinge-loss which lead to improved clustering during inference, *i.e.* they propose to penalize close proximity of the mean embedding of different objects. (Novotny et al., 2018) later showed that constructing dense pixel embeddings to separate objects is not possible with a fully convolutional setup.

EMBEDSEG uses a branched ERF-Net (Romera et al., 2018; Neven et al., 2019), such that each pixel $\boldsymbol{x}_i \in S_k$, in an object instance with label $k$, is trained to predict $(i)$ an offset vector $\boldsymbol{o}_i$ that embeds $\boldsymbol{x}_i$ to $\boldsymbol{e}_i = \boldsymbol{x}_i + \boldsymbol{o}_i$, ideally coinciding with a uniquely defined embedding location $\boldsymbol{e}_i^k$ for the ground truth mask $S_k$, $(ii)$ an uncertainty vector $\boldsymbol{\sigma_i}$ that estimates the error of $\boldsymbol{e}_i$ w.r.t. $\boldsymbol{e}_i^k$, and $(iii)$ a *seediness score* $s_i$ that expresses the likelihood that this pixel coincides with $\boldsymbol{e}_i^k$. Interestingly, the loss terms that enable the training of these

---

1. A memory-efficient open-source implementation of EMBEDSEG is available on GitHub.

2. Data download links can be found on GitHub as well.
   (https://github.com/juglab/EmbedSeg)

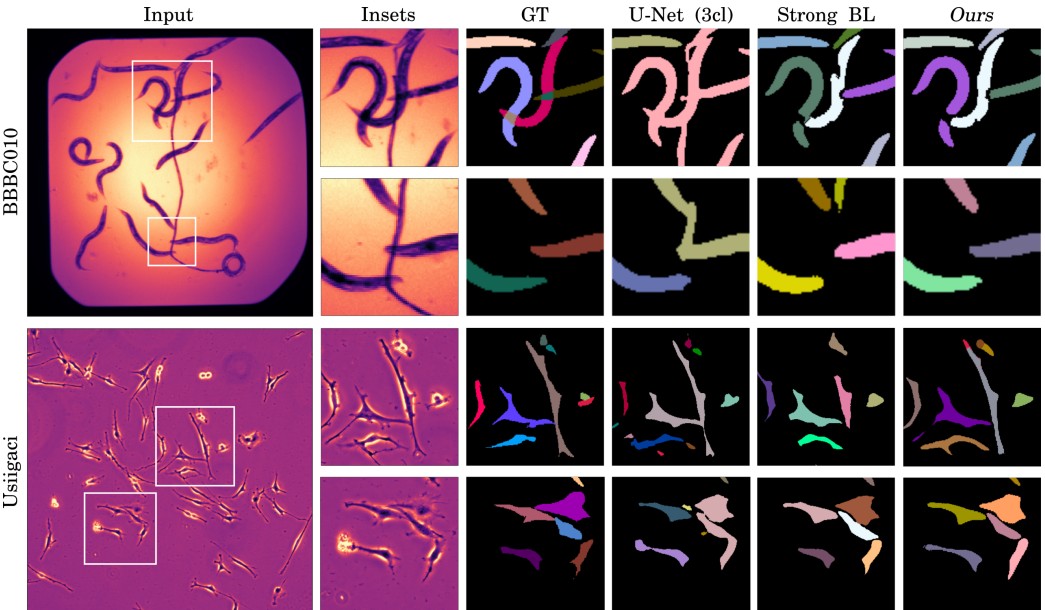

Figure 1: Qualitative results. EMBEDSEG and two baselines compared on representative images of the BBBC010 and Usiigaci datasets. Columns show: full input image, zoomed insets, ground truth labels (GT), and instance segmentation results by the 3-class U-Net baseline, the best performing competing baseline, and EMBEDSEG. Note that each segmented instance is shown in a unique random color.

predicted values also ensures that the IoU of $S_k$ and the predicted instance segmentation is maximized. Additional details are provided in Appendix A.

Once trained, the following inference scheme is used to find object instances (see Appendix B for more details): ($i$) we collect all pixels with a seediness score $s_i > s_{\text{fg}}$ in a set of foreground pixels $S_{\text{fg}}$, ($ii$) from all pixels in $S_{\text{fg}}$, we pick $\boldsymbol{x}_{\text{seed}}$, the pixel with the highest seediness score $s_i > s_{\text{min}}$, ($iii$) if such a $\boldsymbol{x}_{\text{seed}}$ exists, we collect all foreground pixels in $S_{fg}$ that embed themselves at a location where the embedding likelihood defined by $\boldsymbol{e}_{\text{seed}}$ and $\boldsymbol{\sigma}_{\text{seed}}$ is $> 0.5$. Together, these pixels define a segmented instance $S_k$. Finally, ($iv$) we remove all pixels $S_k$ from $S_{\text{fg}}$ and jump to step two until no more valid seed pixels $\boldsymbol{x}_{\text{seed}}$ exist in $S_{\text{fg}}$. In all our experiments we use $s_{\text{fg}} = 0.5$ and $s_{\text{min}} = 0.9$.

While Neven *et al.* either learn the desired embedding location during training or simply use the centroid, we argue that this is not the optimal choice when object shapes are more complex (*i.e.* not star-convex). We reason that it is desirable to choose a point that minimizes the average distance to all pixels $\boldsymbol{x}_i \in S_k$, *i.e.* the *geometric median* (GM). Like the centroid, also the GM has the unfortunate property that it can lie outside of its defining object. Such object-external points are bad embedding points for two reasons: ($i$) the seediness score of such points will likely be very low, and ($ii$) multiple such points might fall very close to each other in crowded image regions. Hence, we propose to use the *medoid* instead. The medoid pixel of the object instance $S_k$ is the one pixel of the object with the smallest average distance to all other pixels *i.e.* $\boldsymbol{x}_{\text{medoid}}(S_k) = \arg\min_{\boldsymbol{y} \in S_k} \frac{1}{|S_k|} \sum_{\boldsymbol{x} \in S_k} \|\boldsymbol{x}, \boldsymbol{y}\|_2$.

Table 1: Quantitative evaluation on three 2D datasets. For each dataset, we compare results of multiple baselines (rows) to results obtained with our proposed pipeline (EmbedSeg) highlighted in gray. First results column shows the required GPU-memory (training) of the respective method. The remaining columns show the Mean Average Precision ($AP_{dsb}$, see main text) for selected IoU thresholds. Best and second best performing methods per column are indicated in bold and underlined, respectively.

| | $GPU_{GB}$ | $AP_{0.50}$ | $AP_{0.55}$ | $AP_{0.60}$ | $AP_{0.65}$ | $AP_{0.70}$ | $AP_{0.75}$ | $AP_{0.80}$ | $AP_{0.85}$ | $AP_{0.90}$ |
|---|---|---|---|---|---|---|---|---|---|---|
| | | | | | *BBBC010* | | | | | |
| 3-Class Unet | 5.6 | 0.521 | 0.466 | 0.451 | 0.440 | 0.427 | 0.407 | 0.377 | 0.332 | 0.243 |
| Cellpose (public) | | 0.225 | 0.204 | 0.184 | 0.155 | 0.097 | 0.043 | 0.013 | 0.002 | 0.000 |
| Harmonic Emb. | | 0.900 | | | | | 0.723 | | | |
| PatchPerPix | | 0.930 | | 0.905 | | 0.879 | | **0.792** | | **0.386** |
| Neven *et al.* | <1 | 0.953 | 0.941 | 0.927 | 0.904 | 0.878 | 0.830 | 0.731 | 0.563 | 0.297 |
| EmbedSeg (*Ours*) | <1 | **0.965** | **0.954** | **0.934** | **0.917** | **0.896** | **0.854** | 0.762 | **0.596** | 0.326 |
| | | | | | *Usiigaci* | | | | | |
| 3-Class Unet | 5.6 | 0.245 | 0.188 | 0.133 | 0.090 | 0.049 | 0.016 | 0.008 | 0.000 | 0.000 |
| Cellpose (public) | | 0.291 | 0.237 | 0.169 | 0.128 | 0.066 | 0.031 | 0.010 | 0.000 | 0.000 |
| Cellpose (*Usiigaci*) | 3.6 | **0.704** | 0.600 | 0.499 | 0.370 | 0.258 | 0.138 | 0.040 | 0.005 | 0.000 |
| Mask R-CNN | 6.9 | 0.583 | 0.520 | 0.439 | 0.365 | 0.235 | 0.130 | **0.045** | **0.008** | 0.000 |
| StarDist | 6.9 | 0.510 | 0.427 | 0.337 | 0.235 | 0.143 | 0.076 | 0.019 | 0.002 | 0.000 |
| Neven *et al.* | 2.9 | 0.648 | 0.570 | 0.463 | 0.343 | 0.233 | 0.115 | 0.035 | 0.004 | 0.000 |
| EmbedSeg (*Ours*) | 2.9 | **0.704** | **0.643** | **0.535** | **0.414** | **0.273** | **0.140** | 0.044 | 0.005 | 0.000 |
| | | | | | *DSB* | | | | | |
| 3-Class Unet | 5.6 | 0.806 | 0.775 | 0.743 | 0.701 | 0.654 | 0.578 | 0.491 | 0.374 | 0.226 |
| Cellpose (public) | | 0.868 | 0.852 | 0.829 | 0.802 | 0.755 | 0.676 | 0.563 | 0.418 | 0.234 |
| Cellpose (*DSB*) | 3.6 | 0.853 | 0.826 | 0.812 | 0.792 | **0.768** | **0.716** | **0.645** | **0.536** | **0.402** |
| Mask R-CNN | 6.9 | 0.832 | 0.805 | 0.773 | 0.730 | 0.684 | 0.597 | 0.489 | 0.353 | 0.189 |
| PatchPerPix | | 0.868 | | 0.827 | | 0.755 | | 0.635 | | 0.379 |
| StarDist | 6.9 | 0.864 | 0.836 | 0.804 | 0.755 | 0.685 | 0.586 | 0.450 | 0.287 | 0.119 |
| Neven *et al.* | 1.3 | 0.873 | 0.852 | 0.830 | 0.799 | 0.762 | 0.704 | 0.623 | 0.511 | 0.373 |
| EmbedSeg (*Ours*) | 1.3 | **0.876** | **0.858** | **0.834** | **0.806** | **0.768** | 0.715 | **0.645** | 0.530 | 0.399 |

During prediction we use 8-fold and 16-fold test-time augmentation in 2D and 3D, respectively (Zeng et al., 2017; Wang et al., 2019) where the evaluation images are transformed through axis-aligned rotations and flips, their corresponding predictions are back transformed and averaged.

## 3. Baselines, Experiments and Results

We measure the performance of EmbedSeg against several state-of-the-art baseline methods that have been developed for microscopy instance segmentation. For 2D images, we tested all methods on three publicly available datasets, namely the *BBBC010 C. elegans* brightfield dataset (Ljosa et al., 2012)[3], the *Usiigaci* NIH/3T3 phase-contrast dataset (Tsai et al., 2019), and the *DSB* data from the Kaggle Data Science Bowl challenge

---

3. We used the *C. elegans* infection live/dead image set version 1 provided by Fred Ausubel and available from the Broad Bioimage Benchmark Collection

Table 2: Quantitative Evaluation on four 3D datasets. For each dataset, we compare results of multiple baselines (rows) to results obtained with our proposed pipeline (EMBEDSEG) highlighted in gray. First results column shows the required GPU-memory (training) of the respective method. The remaining columns show the Mean Average Precision ($AP_{dsb}$) for selected IoU thresholds. Best and second best performing methods per column are indicated in bold and underlined, respectively.

| | $GPU_{GB}$ | $AP_{0.1}$ | $AP_{0.2}$ | $AP_{0.3}$ | $AP_{0.4}$ | $AP_{0.5}$ | $AP_{0.6}$ | $AP_{0.7}$ | $AP_{0.8}$ | $AP_{0.9}$ |
|---|---|---|---|---|---|---|---|---|---|---|
| *Mouse-Organoid-Cells-CBG* | | | | | | | | | | |
| Cellpose (Mouse-Organoid-Cells-CBG) | 3.6 | 0.217 | 0.214 | 0.212 | 0.210 | 0.203 | 0.197 | 0.183 | 0.146 | 0.042 |
| StarDist-3D | 20 | **0.988** | **0.982** | **0.982** | **0.982** | **0.973** | 0.970 | 0.958 | 0.774 | 0.052 |
| EMBEDSEG (Ours) | 7 | **0.988** | **0.982** | **0.982** | **0.982** | **0.973** | **0.973** | **0.973** | **0.970** | **0.929** |
| *Platynereis-Nuclei-CBG* | | | | | | | | | | |
| Cellpose (*Platynereis*-Nuclei-CBG) | 3.6 | 0.971 | 0.971 | 0.966 | 0.957 | 0.931 | 0.872 | 0.700 | 0.299 | **0.009** |
| StarDist-3D | 20 | 0.973 | 0.969 | 0.966 | 0.966 | 0.937 | 0.910 | 0.736 | 0.246 | 0.002 |
| EMBEDSEG (Ours) | 7 | **0.982** | **0.982** | **0.982** | **0.975** | **0.964** | **0.932** | **0.804** | **0.361** | 0.004 |
| *Mouse-Skull-Nuclei-CBG* | | | | | | | | | | |
| Cellpose (Mouse-Skull-Nuclei-CBG) | 3.6 | 0.613 | 0.587 | 0.587 | 0.563 | 0.515 | 0.471 | 0.389 | 0.316 | **0.064** |
| StarDist-3D | 20 | 0.468 | 0.468 | 0.400 | 0.358 | 0.264 | 0.138 | 0.034 | 0.000 | 0.000 |
| EMBEDSEG (Ours) | 7 | **0.837** | **0.837** | **0.837** | **0.837** | **0.795** | **0.646** | **0.549** | **0.362** | 0.053 |
| *Platynereis-ISH-Nuclei-CBG* | | | | | | | | | | |
| Cellpose (*Platynereis*-ISH-Nuclei-CBG) | 3.6 | 0.731 | 0.674 | 0.629 | 0.554 | 0.493 | 0.390 | 0.247 | 0.038 | 0.000 |
| StarDist-3D | 20 | 0.599 | 0.587 | 0.545 | 0.442 | 0.280 | 0.114 | 0.010 | 0.000 | 0.000 |
| EMBEDSEG (Ours) | 7 | **0.884** | **0.884** | **0.874** | **0.852** | **0.781** | **0.655** | **0.482** | **0.120** | 0.000 |

Table 3: Used 3D datasets. In this work we introduce four new volumetric microscopy datasets, covering various practically relevant imaging conditions and microscopy modalities. All datasets come with high quality ground truth labels for training and are publicly available at https://github.com/juglab/EmbedSeg.

| Name | Description | Pixel Size (Z,Y,X) $[\mu m^3]$ | Bit Depth | Used Microscope |
|---|---|---|---|---|
| Mouse-Organoid-Cells-CBG | Mouse Embryonic Stem Cells, R1 cell line, labeled membrane | (1.0, 0.1733, 0.1733) | uint16 | Selective Plane Illumination Microscopy |
| *Platynereis*-Nuclei-CBG | Nuclei of a developing *Platynereis dumerilii* embryo at stages between 0 to 16 hours post fertilization, injected with a fluorescent nuclear tracer | (2.031, 0.406, 0.406) | uint16 | Simultaneous Multi-view Light-Sheet Microscopy |
| Mouse-Skull-Nuclei-CBG | Nuclei of the skull region of developing mouse embryos, labeled with DAPI | (0.200, 0.073, 0.073) | uint16 | Inverted Zeiss LSM 880 Microscope |
| *Platynereis*-ISH-Nuclei-CBG | Nuclei of whole-mount *Platynereis dumerilli* specimens at stage of 16 hours post fertilization, labeled with DAPI | (0.4501, 0.4499, 0.4499) | uint8 | Laser Scanning Confocal Microscopy |

of 2018 (Caicedo et al., 2019a)[4]. For volumetric images, we tested all methods on four new datasets (*Mouse-Organoid-Cells-CBG*, *Platynereis-Nuclei-CBG*, *Mouse-Skull-Nuclei-CBG*, and *Platynereis-ISH-Nuclei-CBG*), which we make available with publishing this work. Additional details can be found in Table 3.

**Chosen Baseline Methods.** *Cellpose* (Stringer et al., 2020) is a spatial-embedding based instance segmentation method where the task of the network is to predict a flow at each

---

4. We used a subset of the image set BBBC038v1, available from the Broad Bioimage Benchmark Collection

pixel. This ground truth vector flow field is pre-computed from the instance masks as solution to the heat diffusion equation, assuming a heat source placed at the center of the object instance. These learnt flows are followed, during inference, to group pixels which arrive at the same location. *PatchPerPix* (Hirsch et al., 2020) is a method that predicts a dense binary mask per pixel. These learnt local per-pixel (per-voxel) shape descriptor masks are, during inference, assembled into complete object instances. *StarDist* (Schmidt et al., 2018) and *StarDist-3D* (Weigert et al., 2020) are recently the arguably most widely applied methods in microscopy image analysis. StarDist predicts at each pixel (voxel) the distance to the boundary (outline) of the surrounding object along a given set of directions (rays). A *3-Class Unet* (Ronneberger et al., 2015) is another widely adopted method for semantic segmentation, *i.e.* the assignment of one of three classes (background, foreground, border) to each pixel (voxel). During inference, pixels (voxels) of a given class are typically clustered into instance segmentations by finding connected components.

Cellpose, next to offering code for training, also offers a public model, trained on a huge and diverse set of training data. Hence, below we report not only the performance of Cellpose trained on each dataset individually, but also how well the public model performs (see Table 1).

**Data and Data Handling in 2D.** The *BBBC010 dataset* consist of only 100 images of $696 \times 520$ pixels each. Like others before us, we randomly split these images in two equally sized sets, one used for training, the other to evaluate performance (testing). We cropped $256 \times 256$ patches that are centered around each ground truth object (worm) and have used 15% of all crops as validation set. Reported results are averages over 9 independent data-splits and training runs. For the *Usiigaci dataset*, we split the 50 images of size $1024 \times 1022$ pixels as suggested by Tsai *et al.* (Tsai et al., 2019) in 45 training and 5 test images. We cropped $512 \times 512$ patches that are centered on all ground truth objects. The *DSB dataset* is the largest collection of images, of which we use the same subset as originally suggested in (Schmidt et al., 2018). It contains a total of 497 images of variable size and is pre-split in 447 training and 50 test images. We train on object-centered $256 \times 256$ crops. For the DSB and Usiigaci datasets, we hold out 15 % of all training images chosen randomly for validation purposes, prior to cropping, and also average results over 9 independent runs.

**Data and Data Handling in 3D.** The *Mouse-Organoid-Cells-CBG dataset* is the largest collection of 3D images, consisting of 108 volumes of $70 \times 378 \times 401$ (Z, Y, X) voxels each. We randomly select 15 and 11 images for validation and testing, respectively. Training is performed on object-centered crops of size $32 \times 200 \times 200$. The *Platynereis-Nuclei-CBG dataset* contains 9 images ($113 \times 660 \times 700$ voxels each), of which we randomly select 2 and 2 images for validation and testing, respectively. Training is performed on object-centered crops of size $32 \times 136 \times 136$. The *Mouse-Skull-Nuclei-CBG dataset* contains only 2 images of $209 \times 512 \times 512$ and $125 \times 512 \times 512$ voxels respectively. Due to very limited amount of available data, we test on the sub-volume $(:, :, 256:512)$ of the second image. Training is performed on the remaining data using object-centered crops of size $96 \times 128 \times 128$. The *Platynereis-ISH-Nuclei-CBG dataset* also contains 2 images of $515 \times 648 \times 648$ voxels each. We test the performance on the the sub-volume $(300:405, :, :)$ of the second image and train on object-centered crops of size $80 \times 80 \times 80$ on the remaining data.

For all 3D datasets, we report the average results on the test data over 3 independent runs.

**Training Details.** All results obtained with EMBEDSEG and the method by Neven *et al.* on 2D datasets use the Branched ERF-Net (Romera et al., 2018; Neven et al., 2019) architecture, the Adam optimizer (Kingma and Ba, 2014) with a decaying learning rate $\alpha_i = 5e^{-4}\left[1 - \frac{i}{200}\right]^{0.9}$, where $i$ denotes the current epoch. For training and inference on 3D datasets, we propose a Branched ERF-Net operating on 3D convolutions (see Appendix E for a schematic of our proposed architecture).

For the BBBC010 data, we use a batch size of 1 without virtual batch multiplier, while for other datasets we employ a batch-size of 2 and a virtual batch multiplier of 8 (giving us an effective batch-size of 16).

During training, axis-aligned rotations and flips were used for augmenting the available data. Every training was run for 200 epochs, and the model with the best performance w.r.t. IoU on the validation data is later used for reporting results on the evaluation data (see Tables 1 and 2).

**Performance Evaluations.** All results on 2D images are compared using the Mean Average Precision ($AP_{dsb}$ score (Schmidt et al., 2018)), at IoU thresholds ranging from 0.5 to 0.9 (see Table 1), while the results on volumetric images are evaluated on at IoU thresholds ranging from 0.1 to 0.9 (see Table 2). For all EMBEDSEG and Neven *et al.* results, we compute the minimum object size in terms of the number of interior pixels using the available training and validation masks. We then use this value during inference to avoid spurious false positives.

**Ablation Studies.** In order to evaluate the contribution of ($i$) using the medoid instead of the centroid in EMBEDSEG, and ($ii$) employing test-time augmentation, we have performed the respective ablation studies and report the results on two 2D and one 3D dataset in Table 4.

## 4. Discussion

In this work we propose EMBEDSEG, an embedding-based instance segmentation method for 2D and 3D microscopy data, inspired by the work of (Newell et al., 2017; Brabandere et al., 2017; Neven et al., 2019) and others. The unmodified[5] embedding-based method by Neven *et al.* shows promising results on 2D microscopy data, but the modifications we propose (medoid embedding, test-time augmentation, extension to 3D, hyper-params deduced from training data, one-hot encoded masks, *etc.*) secure EMBEDSEG's state-of-the-art results on many practically relevant biomedical microscopy datasets in two and three dimensions.

When comparing the results of EMBEDSEG to all obtained baseline predictions we noticed that Cellpose often runs into issues in dense 3D regions. For example, Cellpose results on the Mouse-Organoid-Cells-CBG dataset produce a lot of spurious over-segmentations, which we believe are a side-effect of Cellpose's interpolation of individual 2D predictions of the sliced 3D input.

---

5. Small adaptions of the code by Neven *et al.* (Neven et al., 2019) are required in order to deal with non-RGB images, one hot encoded instance masks etc.

Table 4: Ablation studies. For the BBBC010, Usiigaci and *Platynereis*-Nuclei-CBG datasets, we show how using the centroid instead of the medoid during training and/or removing test-time augmentation negatively impacts overall performance. Like before, columns show $AP_{dsb}$ (first row) or $\Delta AP_{dsb}$ (rows 2-4) at selected IoU thresholds. Individual rows show: results obtained with EMBEDSEG; using centroids instead of medoids during training; using medoids but without test-time augmentation; using centroids during training and no test-time augmentation.

| $AP_{dsb}$ | $AP_{0.50}$ | $AP_{0.55}$ | $AP_{0.60}$ | $AP_{0.65}$ | $AP_{0.70}$ | $AP_{0.75}$ | $AP_{0.80}$ | $AP_{0.85}$ | $AP_{0.90}$ |
|---|---|---|---|---|---|---|---|---|---|
| *BBBC010* (2D) | | | | | | | | | |
| EMBEDSEG (ours) | 0.965 | 0.954 | 0.934 | 0.917 | 0.896 | 0.854 | 0.762 | 0.596 | 0.326 |
| ↳ medoid ⇒ centroid | -0.002 | -0.002 | -0.000 | -0.002 | -0.001 | -0.004 | +0.004 | +0.001 | +0.003 |
| ↳ no test-time augm. | -0.007 | -0.008 | -0.003 | -0.008 | -0.014 | -0.020 | -0.028 | -0.033 | -0.025 |
| ↳ both (=Neven *et al.*) | -0.011 | -0.013 | -0.007 | -0.012 | -0.018 | -0.024 | -0.032 | -0.033 | -0.029 |
| *Usiigaci* (2D) | | | | | | | | | |
| EMBEDSEG (ours) | 0.704 | 0.643 | 0.535 | 0.414 | 0.273 | 0.140 | 0.044 | 0.005 | 0.000 |
| ↳ medoid ⇒ centroid | -0.014 | -0.013 | -0.006 | -0.005 | +0.006 | +0.009 | +0.002 | -0.001 | 0.000 |
| ↳ no test-time augm. | -0.028 | -0.048 | -0.050 | -0.052 | -0.040 | -0.030 | -0.008 | -0.001 | 0.000 |
| ↳ both (=Neven *et al.*) | -0.038 | -0.053 | -0.053 | -0.055 | -0.028 | -0.020 | -0.006 | 0.000 | 0.000 |
| $AP_{dsb}$ | $AP_{0.10}$ | $AP_{0.20}$ | $AP_{0.30}$ | $AP_{0.40}$ | $AP_{0.50}$ | $AP_{0.60}$ | $AP_{0.70}$ | $AP_{0.80}$ | $AP_{0.90}$ |
| *Platynereis*-Nuclei-CBG (3D) | | | | | | | | | |
| EMBEDSEG (ours) | 0.982 | 0.982 | 0.982 | 0.975 | 0.964 | 0.932 | 0.804 | 0.361 | 0.004 |
| ↳ medoid ⇒ centroid | -0.006 | -0.006 | -0.008 | -0.005 | -0.010 | -0.007 | +0.018 | +0.026 | 0.000 |
| ↳ no test-time augm. | -0.012 | -0.012 | -0.012 | -0.012 | -0.013 | -0.019 | -0.023 | -0.037 | -0.003 |
| ↳ both | -0.013 | -0.014 | -0.016 | -0.018 | -0.022 | -0.013 | -0.033 | -0.019 | 0.000 |

StarDist-3D does not have this problem, but is naturally challenged when the objects to be segmented are not star-convex (*e.g.* for the Mouse-Skull-Nuclei-CBG and *Platynereis*-ISH-Nuclei-CBG datasets). On datasets that contain only star-convex objects, *e.g.* labeled cell nuclei, StarDist-3D typically performs on-par or even better than EMBEDSEG (see Appendix C for an example).

Additionally, we noticed that StarDist-3D performance generally drops at higher IoU-thresholds ($AP_{\geqslant 0.7}$ in Tables 1 and 2). When we analyzed the reason for this, we found that this is caused by the planarity of faces defined by the predicted vectors that span the star-convex object instances[6].

An additional and practically very relevant advantage of EMBEDSEG is its small memory footprint on the GPU, even during training, see Tables 1 and 2. This can enable users to benefit from our method even on cheap laptop hardware. Hence, we strongly feel that the method we propose will lead to improved instance segmentations in in many biomedical projects that require the analysis of microscopy data in two or three dimensions.

---

6. While object instances typically have smooth, rounded surfaces, StarDist-3D instances are defined by the convex hull of a given number of vectors radiating out of a source pixel. Such a linear shape approximation causes lower IoU-values and are therefore leading to weaker AP scores at high IoU-thresholds

## Acknowledgments

The authors would like to thank the Scientific Computing Facility at MPI-CBG, thank Matthias Arzt, Joran Deschamps and Nuno Pimpão Martins for feedback and testing. Alf Honigmann and Anna Goncharova provided the Mouse-Organoid-Cells-CBG data and annotations. Jacqueline Tabler and Diana Afonso provided the Mouse-Skull-Nuclei-CBG dataset and annotations. This work was supported by the German Federal Ministry of Research and Education (BMBF) under the codes 031L0102 (de.NBI) and 01IS18026C (ScaDS2), and the German Research Foundation (DFG) under the code JU3110/1-1(FiSS) and TO563/8-1 (FiSS). P.T. was supported by the European Regional Development Fund in the IT4Innovations national supercomputing center, project number CZ.02.1.01/0.0/0.0/16_013/0001791 within the Program Research, Development and Education.

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

## Appendix A. Details on Training EmbedSeg in 2D and 3D

The goal of instance segmentation is to cluster a set of pixels $\boldsymbol{X} = \{\boldsymbol{x}_1 \ldots \boldsymbol{x}_i \ldots \boldsymbol{x}_N\}$, (where $\boldsymbol{x} \in \mathcal{R}^D$, with $D$ being the dimensionality of the given input images), into a set of segmented object instances $S = \{S_1 \ldots S_k \ldots S_K\}$.

This is achieved by learning an offset vector $\boldsymbol{o}_i$ for each pixel $\boldsymbol{x}_i$, so that the resulting (spatial) embedding $\boldsymbol{e}_i = \boldsymbol{x}_i + \boldsymbol{o}_i$ points to its corresponding object center (instance center) $\boldsymbol{C}_k$. Here, $\boldsymbol{o}_i$, $\boldsymbol{e}_i$ and $\boldsymbol{C}_k$ are in $\mathcal{R}^D$.

In order to do so, we propose to use a Gaussian function $\phi_k$ for each object $S_k$, which converts the distance between a (spatial) pixel embedding $\boldsymbol{e}_i$ and the instance center $\boldsymbol{C}_k$ into a probability of belonging to that object

$$\phi_k\left(\boldsymbol{e}_i\right) = \exp\left(-\left\|\frac{\left(\boldsymbol{e}_i - \boldsymbol{C}_k\right)^T \boldsymbol{\Sigma}_k^{-1}\left(\boldsymbol{e}_i - \boldsymbol{C}_k\right)}{2}\right\|\right). \tag{1}$$

A high probability signifies that the pixel embedding $\boldsymbol{e}_i$ is close to the instance center $\boldsymbol{C}_k$ and the corresponding pixel is likely to belong to the object $S_k$, while a low probability means that the pixel is more likely to belong to the background (or another object). More specifically, if $\phi_k(\boldsymbol{e}_i) > 0.5$, the pixel at location $\boldsymbol{x}_i$ will be assigned to the object $S_k$. Here, $\boldsymbol{\Sigma}_k \in \mathcal{R}^{D \times D}$ is the diagonal covariance matrix representing the cluster bandwidth for object $S_k$. The corresponding standard deviation vector for object $S_k$ is indicated as $\boldsymbol{\sigma}_k \in \mathcal{R}^D$ whose entries along the $d^{\text{th}}$ dimension are denoted as $\sigma_{k,d}$. For example, for D = 3,

$$\boldsymbol{\Sigma}_k = \begin{bmatrix} \sigma_{k,1}^2 & 0 & 0 \\ 0 & \sigma_{k,2}^2 & 0 \\ 0 & 0 & \sigma_{k,3}^2 \end{bmatrix}. \tag{2}$$

In order to allow larger objects to predict a larger and similarly, smaller objects to predict a smaller $\boldsymbol{\Sigma_k}$, we let each pixel $\boldsymbol{x}_i$ of object $k$ individually predict a $\boldsymbol{\sigma_i}$ and compute the corresponding $\boldsymbol{\sigma_k}$ for the constituting object as the mean of all predicted $\boldsymbol{\sigma_i}$ for that object

$$\boldsymbol{\sigma_k} = \frac{1}{|S_k|} \sum_{\boldsymbol{\sigma_i} \in S_k} \boldsymbol{\sigma_i}. \tag{3}$$

By comparing the predicted $\phi_k$ of object to the ground truth foreground mask $S_k$, we compute the differentiable Lovász-Softmax loss $L_{\text{IoU}}$ (Berman et al., 2018; Yu and Blaschko, 2015).

There is still the question of deducing the centre of attraction of an object, at inference time, so as to look for pixel embeddings which fall in a *margin* around it. For this purpose, we also let each pixel predict a *seediness* score which indicates how likely it is to be the centre of attraction. The seediness score should actually be similar to the output of the gaussian function in Equation (1). So we can construct a loss function

$$L_{\text{seed}} = \frac{1}{N} \sum_{i=1}^{N} w_{\text{fg}} \mathbb{1}_{\{s_i \in S_k\}} \|s_i - \phi_k(\boldsymbol{e}_i)\|^2 + w_{\text{bg}} \mathbb{1}_{\{s_i \notin S_{\text{fg}}\}} \|s_i - 0\|^2, \tag{4}$$

which allows minimizing the distance between the output of the gaussian function corresponding to any pixel and the predicted seediness score, arising from that pixel. The seediness score for the background pixels are regressed to 0. Furthermore, to ensure that at inference, while sampling highly seeded pixels, $\boldsymbol{\sigma}_k \approx \hat{\boldsymbol{\sigma}}_k$, we include a smoothness loss

$$L_{\text{var}} = \frac{1}{|S_k|} \sum_{\boldsymbol{\sigma}_i \in S_k} \|\boldsymbol{\sigma}_i - \boldsymbol{\sigma}_k\|^2. \tag{5}$$

The complete loss function is then computed as the weighted sum

$$L = w_{\text{seed}}L_{\text{seed}} + w_{\text{IoU}}L_{\text{IoU}} + w_{\text{var}}L_{\text{var}}. \tag{6}$$

We use $w_{\text{seed}} = 1$, $w_{\text{iou}} = 1$ and $w_{\text{var}} = 10$ for all 2D and 3D experiments. For all 2D experiments, we additionally set $w_{\text{fg}}$ and $w_{\text{bg}}$ to 10 and 1, respectively. For all 3D experiments, $w_{\text{fg}}$ was instead set to the ratio of the number of background and foreground pixels in training and validation data. More details can be found in (Neven et al., 2019) and in our open source implementation at https://github.com/juglab/EmbedSeg.

## Appendix B. Visualizing the Inference Process of EmbedSeg

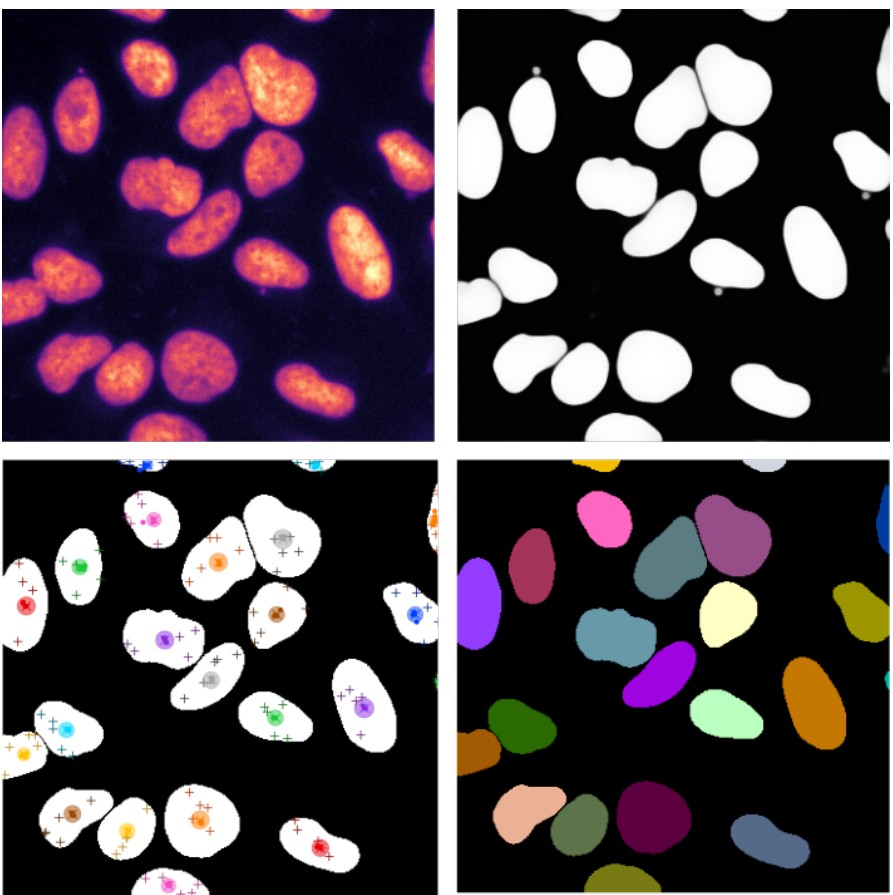

Figure B.1: **Visualization of inference procedure of EmbedSeg.** An exemplary input image (top-left), we iteratively pick seed pixels greedily from the predicted seediness map (top-right) and cluster other foreground pixels w.r.t. their predicted embeddings as explained in Section 2. In the bottom left image we show: ground truth instances as a binary mask (white regions), the embedding location and clustering bandwidth (thresholded at a likelihood of 0.5) of iteratively picked seed pixels (larger, semi-transparent ellipses), and the learnt spatial embedding locations (smaller dots inside ellipses) of 5 randomly chosen foreground pixels per predicted instance (colored plus signs). The final predicted instance segmentation result is shown in the bottom-right panel.

Figure B.1 gives a behind-the-scenes look at the process of clustering pixels into object instances. Please also consult our open GitHub repository for more visualizations and details (https://github.com/juglab/EmbedSeg).

## Appendix C. Results on a 3D Dataset Containing Star-Convex Object Instances

The *Paryhale*-Nuclei-IGFL data (Alwes et al., 2016) is a dataset which was used to demonstrate the performance of Stardist-3D (Weigert et al., 2020). It contains a total of 6 images of $34 \times 512 \times 512$ (Z, Y, X) voxels each. Using EMBEDSEG, we train on object-centred $24 \times 120 \times 120$ crops. We randomly put aside 1 image for evaluation, and then hold out 2 training images chosen randomly for testing (performance evaluation). Results are averaged over 3 independent runs (one per held out dataset).

Table 5: Quantitative results on the *Paryhale*-Nuclei-IGFL data. We compare results of multiple baselines (rows) to results obtained with our proposed pipeline (EMBEDSEG), highlighted in gray. First results column shows the required GPU-memory (training) of the respective method. The remaining columns show the Mean Average Precision ($AP_{dsb}$, see main text) for selected IoU thresholds. Best and second best performing methods per column are indicated in bold and underlined, respectively.

| | $GPU_{GB}$ | $AP_{0.1}$ | $AP_{0.2}$ | $AP_{0.3}$ | $AP_{0.4}$ | $AP_{0.5}$ | $AP_{0.6}$ | $AP_{0.7}$ | $AP_{0.8}$ | $AP_{0.9}$ |
|---|---|---|---|---|---|---|---|---|---|---|
| *Paryhale-Nuclei-IGFL* (Alwes et al., 2016; Weigert et al., 2020) | | | | | | | | | | |
| U-Net | | 0.592 | 0.552 | 0.481 | 0.372 | 0.280 | 0.198 | 0.097 | 0.010 | 0.000 |
| Cellpose (*Paryhale*-Nuclei-IGFL ) | 3.6 | 0.545 | 0.498 | 0.456 | 0.384 | 0.285 | 0.154 | 0.040 | 0.006 | 0.000 |
| StarDist-3D | 20 | **0.766** | **0.757** | **0.741** | **0.698** | **0.593** | **0.443** | **0.224** | **0.038** | 0.000 |
| EMBEDSEG (Ours) | 7 | 0.581 | 0.581 | 0.579 | 0.543 | 0.472 | 0.359 | 0.185 | **0.038** | 0.000 |

## Appendix D. Qualitative 3D Segmentation Results

Figures D.1 to D.4 show representative qualitative results on all four volumetric microscopy datasets introduced in Table 3.

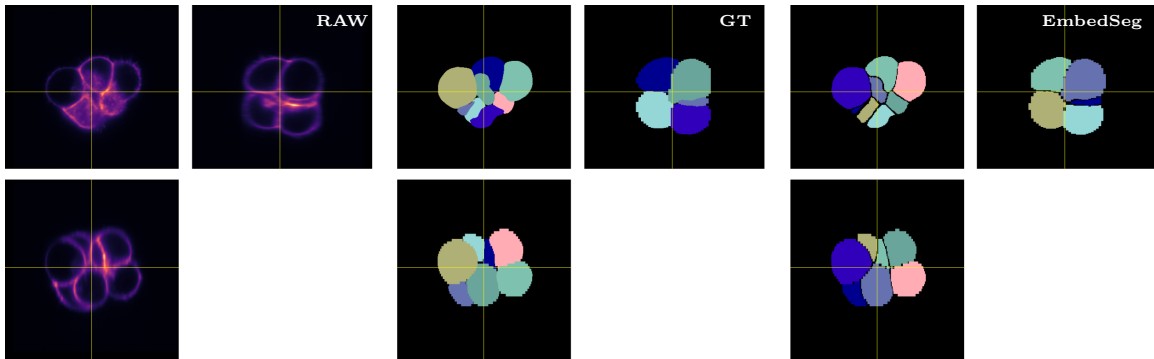

Figure D.1: Qualitative results of EMBEDSEG on the Mouse-Organoid-Cells-CBG dataset. Columns show orthogonal $XY$, $YZ$ and $XZ$ slices of one representative input image, ground truth labels (GT), and our instance segmentation results using EMBEDSEG, respectively. Note that each segmented instance is shown in a random but unique color.

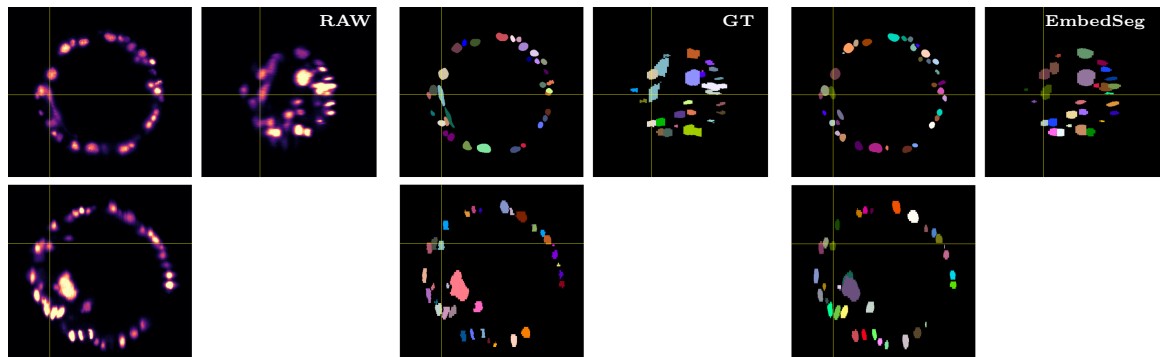

Figure D.2: Qualitative results of EMBEDSEG on the *Platynereis*-Nuclei-CBG dataset. Columns show orthogonal $XY$, $YZ$ and $XZ$ slices of one representative input image, ground truth labels (GT), and our instance segmentation results using EMBEDSEG, respectively. Note that each segmented instance is shown in a random but unique color.

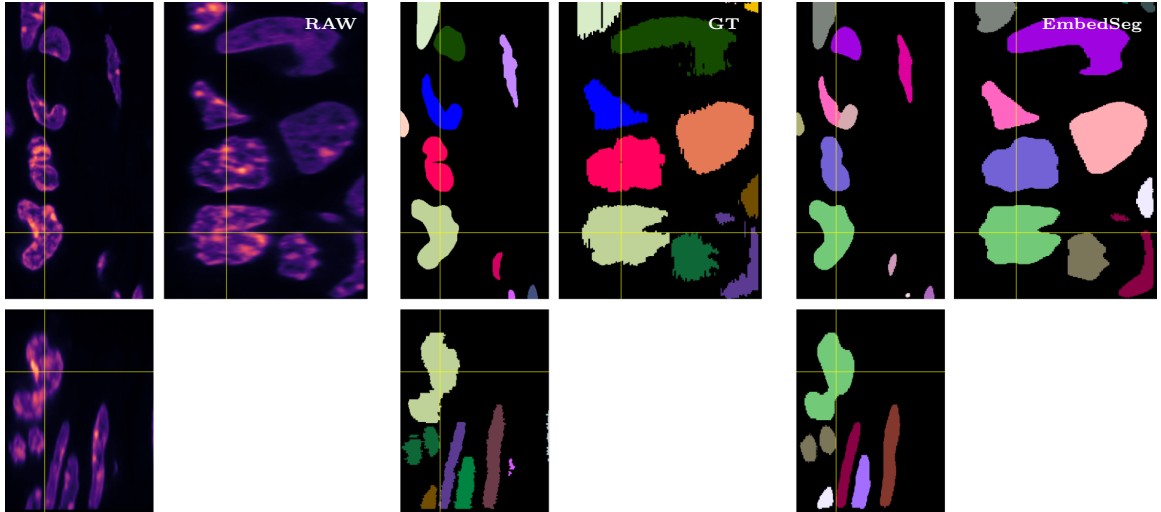

Figure D.3: Qualitative results of EmbedSeg on the Mouse-Skull-Nuclei-CBG dataset. Columns show orthogonal $XY$, $YZ$ and $XZ$ slices of one representative input image, ground truth labels (GT), and our instance segmentation results using EmbedSeg, respectively. Note that each segmented instance is shown in a random but unique color.

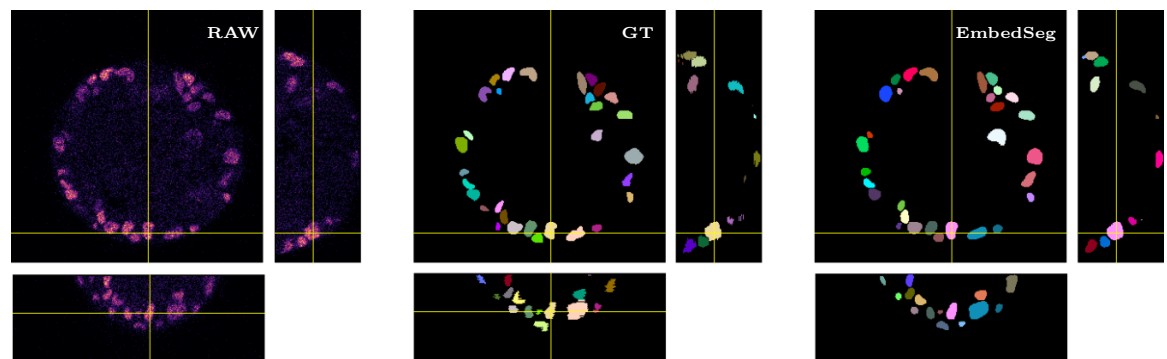

Figure D.4: Qualitative results of EmbedSeg on the *Platynereis*-ISH-Nuclei-CBG dataset. Columns show orthogonal $XY$, $YZ$ and $XZ$ slices of one representative input image, ground truth labels (GT), and our instance segmentation results using EmbedSeg, respectively. Note that each segmented instance is shown in a random but unique color.

## Appendix E. 3D Network Architecture

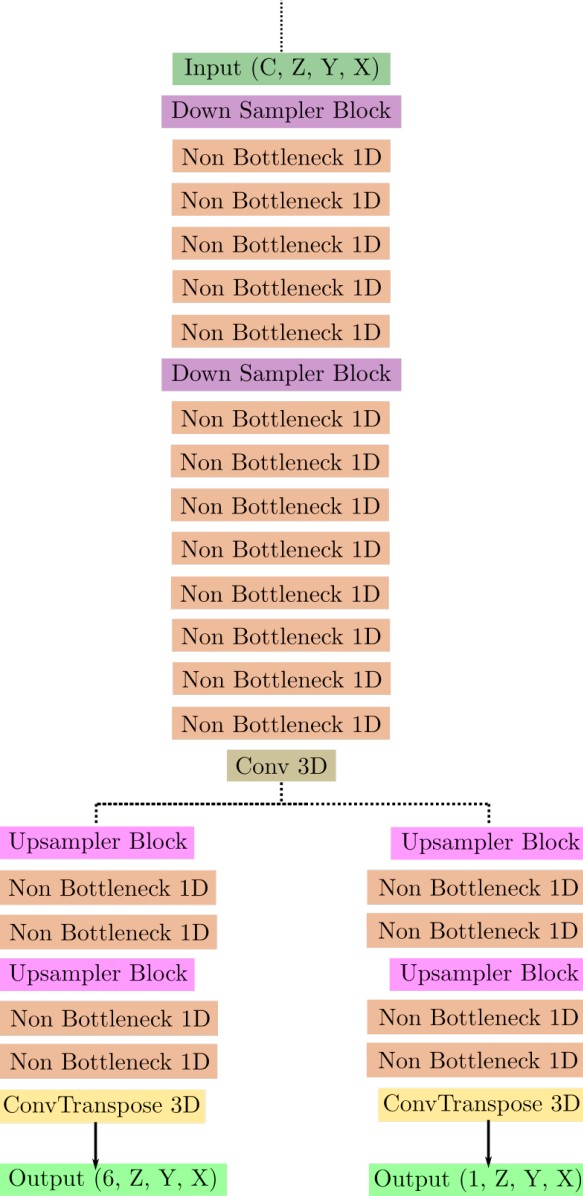

Figure E.1: Schematic representing the architecture of the used 3D Branched ERF-Net which accepts volumetric images with $C$ color channels. The encoder portion of the network has two branches: the first branch returns 6 outputs per pixel which represent the offsets and the clustering bandwidths in x, y and z dimensions. The second branch returns one output per pixel which represents the 'seediness' score of the pixel.

