# OpenReview forum: "Embedding-based Instance Segmentation in Microscopy"
_MIDL.io/2021/Conference — MIDL 2021_

### Official Review · AnonReviewer2 · 2021-03-08

**Confidence:** 3
**Preliminary Rating:** 4
**Recommendation:** Oral, Poster
**Final Rating:** 4

**Summary:**

The paper proposes EmbedSeg, an instance segmentation method based on spatial embeddings for segmenting microscopy images. The paper compares the proposed method to relevant microscopy instance segmentation tools and shows that EmbedSeg achieves comparable or superior performance at a lower memory footprint. An ablation study verifies that EmbedSeg improves upon previous work on embedding-based instance segmentation by using mediods instead of centroids and making use of test-time augmentation.

**Strengths:**

The paper sets out a clear application of segmenting object instances in microscopy images which has seen some interest but has not been fully reliably solved yet. The paper then makes use of recent instance segmentation methods using spatial embeddings and translates those advances to the domain of microscopy imaging. It offers a well-rounded evaluation across different 2D and 3D.

**Weaknesses:**

The paper lacks a little in clarity of related work and background and the used notation is not properly explained. Further, she of the datasets for evaluation are very small and it is not clear how conclusive the results on them can be.

**Deanonymize Review:**

no

**Detailed Comments:**

- What is S_k (mentioned in 2.)?  What's the index k related to?
- How are multiple test-time augmentation predictions combined? Is this majority voting or averaging on the probabilities?
- Why are some methods missing values in the GPU memory column? Also how do the methods relate on train and runtime? Is it actually feasible to train this on laptops or is it only possible to run the trained pipeline?
- For Cellpose it is mentioned that it predicts a flow at each pixel - is this an "optical flow" - might be good to spell this out. Also it might be worthwhile to elaborate on the comparison between EmbedSeg and Cellpose as Cellpose seems to be the closest baseline method (apart from the original ERF-Net).
- How are certain parameters chosen, like discarding objects smaller than 37 px? Is this the same for 2D and 3D data?

**Final Rating Justification:**

I thank the authors for their rebuttal. Most of my points have been addressed and I stick to my previous suggestion of accepting the paper.

**Justification Of The Preliminary Rating:**

The paper presents a strong case for a using spatial embedding-based instance segmentation methods in microscopy imaging. There it makes modifications to previous work improving the performance on this domain and comparing the performance to various baseline methods on a wide range of tasks. It is a good read and well presented.

**Paper Type:**

both

**Special Issue:**

yes

---

> ### Author Response · Authors · 2021-03-17
> **Response to AnonReviewer2**
>
> We thank you for your very meticulous reading  of our submission.
> We also thank you for your comments.
> We address your suggestions one by one below:
>
>
> * *"What is $S_k$ (mentioned in 2.)? What's the index k related to?"*
>
>     + We introduce $S_k$ on page *2* in section *Related Work and Proposed Method*. Also, this is discussed in more detail on page *11* in *Appendix A*. $S_k$ refers to the subset of pixels in the image which are assigned to an object with index/label k.
>
> * *"How are multiple test-time augmentation predictions combined? Is this majority voting or averaging on the probabilities?"*
>
>     + We simply use averaging for combining multiple test-time predictions. We do mention this explicitly now on page *4* in section *Related Work and Proposed Method*.
>
> * *"Why are some methods missing values in the GPU memory column? Also how do the methods relate on train and runtime? Is it actually feasible to train this on laptops or is it only possible to run the trained pipeline?"*
>
>     + We have added numbers in cases when we trained the method ourselves. *Cellpose (public)* was not trained by us, hence the corresponding entry is empty. Numbers for methods such as *PatchPerPix* and *Harmonic Embeddings* were reported from their respective publication. Last but not least, the missing memory footprint for *Neven et al.* was a mistake that is now fixed.
>
>     With respect to you second question, it is indeed feasible to train *EmbedSeg* on laptops (which the authors have done extensively). All reported numbers refer to GPU memory consumption during training. We mention this, for example, in the caption of *Table 1* on have now made this more explicit in the *Discussion* section on page *8*.
>
> * *"For Cellpose it is mentioned that it predicts a flow at each pixel - is this an "optical flow" - might be good to spell this out. Also it might be worthwhile to elaborate on the comparison between EmbedSeg and Cellpose as Cellpose seems to be the closest baseline method (apart from the original ERF-Net)."*
>
>     + We have added some clarifying words on page *6* in Section *Baselines, Experiments and Results*.
>
> * *"How are certain parameters chosen, like discarding objects smaller than 37 px? Is this the same for 2D and 3D data?"*
>
>     + This comment was very helpful for us. It pointed to a hyperparameter which was indeed set to a very specific value. To be honest, we did neglect to care about this parameter in the heat of finishing the manuscript and it simply happened to be $37$.
>     After being called out on this odd choice, we have now changed our thinking about this parameter and updated our overall workflow slightly.
>     More specifically, we now compute the minimum-object-size directly from ground-truth masks given in the train and validation data and then use this value during testing.
>     This procedure is always applicable and removes the burden of thinking about a suitable value from our users.
>     We re-evaluated ALL results and changed them throughout the manuscript (All changes are highlighted in the uploaded rebuttal PDF, see for example *Table 1* on *Page 4*).
>     Of course we have also updated the main text to describe this updated procedure, see for example highlights on page *7* in section *Baselines, Experiments and Results*.
>     Note: numbers are not very different, at times our results are slightly better. Nothing fundamental changed after introducing this update.
>
> * *"The paper lacks a little in clarity of related work and background and the used notation is not properly explained. Further, some of the datasets for evaluation are very small and it is not clear how conclusive the results on them can be."*
>
>     + As mentioned in response to other reviewer comments, we hope that the changes to main text and supplement address your concerns wrt. clarity and notation.
>     Regarding the dataset sizes in 3D we would have preferred them to be larger ourselves. With our biologist collaborators we are, in fact, working on larger 3D data and hope to being able to release these datasets soon. Until then, we would be delighted to be pointed to larger and freely available microscopy datasets.

---

### Official Review · AnonReviewer3 · 2021-03-08

**Confidence:** 5
**Preliminary Rating:** 4
**Recommendation:** Oral
**Final Rating:** 4

**Summary:**

The authors present a method for instance segmentation of microscopy images based on the work of Neven et al., 2019. They introduce some variants to adapt the previous approach to 2D and 3D microscopy image analysis and to reduce the GPU footprint used by their network. Additionally, they test their work on 3 (2D public) and 4 (3D in-house) different datasets, and compare their approach with some state-of-the-art methods, showing the potential of the embedded instance segmentation approaches. The 3D datasets are released within this work.


**Strengths:**

- Instance segmentation in microscopy image analysis is not as explored as in other fields of computer vision. Despite the recent work of StarDist, MaskRCNN, CellPose, or SplineDist [1], there is still much work to do, especially for the cases in which the object density or the shape variability are high. The authors have successfully adapted an existing method in computer vision for instance segmentation to microscopy data and have demonstrated the potential of their approach.

- The code is completely open and free to access. They also provide Jupyter notebooks as examples of data preparation, training, and prediction, which supports the method reproducibility and transfer.

- They contribute with 4 new 3D datasets for instance segmentation.

- The experimental setups are fully documented and they repeat each training schema 9 times to provide averaged accuracies.

[1]Mandal S., Uhlmann V., SplineDist: Automated Cell Segmentation With Spline Curves, bioRxiv 2021, https://www.biorxiv.org/content/10.1101/2020.10.27.357640v2

**Weaknesses:**

- The authors should clean the manuscript and elaborate more on the methodology for its completeness. While most details remain similar to the ones defined in Neven et al., 2017, it is easier to follow the entire description when the parameters and variables are properly described.

- For example, the authors provide visualizations of the seediness scores and embeddings in their GitHub repository. Such visualizations could help to the understanding of their approach and would make clear what is their contribution with respect to the existing work.  This information could be included in the Appendix.

- "The proposed objective encourages pairs of tags to have similar values iff the corresponding pixels belonged to the same object" Does this iff refer to **if and only if**? If so, please make a distinction. If not, please correct it.

- "Greedily pick from all (remaining) pixels the one with the highest predicted seediness score s_i > s_{min}" what does remaining mean?

- It probably makes no difference as it is a matter of scaling inside S_k, but strictly speaking, x_{medoid} should include the 1/|S_k| factor to measure the average distance.

- What is 3-Class referring to in the 3-Class UNet?

- Table 1 and 2: Did you retrain all the methods except for Cellpose (public)? Please, order all the methods following the same order.

- Why didn't you manage to calculate the GPU footprint of Neven et al. method?

- The description of Data and Data Handling in 3D could be improved with minimal effort: when providing the size of the images (e.g. 70 x 378 x 401) I assume the authors are talking about voxels, please indicate it as it was done with 2D data referring to pixels. "we test on the right half of the second image", please indicate the voxel positions as done later with the Platynereis-ISH-Nuclei-CBG dataset.

- In the training details, it is said: "All results obtained with EmbedSeg and the method by Neven et al. use the branched ERF-Net, ... (see Appendix A.4)", but then, in A.4, the shown architecture seems to be for the 3D datasets. Could you please explain this more clearly?

- In the appendix, what are sigma_{k, 1}, sigma_{k, 2} and sigma_{k, 3}? what is \^{sigma}?

- The final loss function (Eq. 6), seems to be a weighted sum. Could you please define w_{seed}, w_{IoU} and w_{var}? Which values would you recommend? Did  you use the same values in all the experiments?

**Deanonymize Review:**

no

**Final Rating Justification:**

I thank the authors for the detailed rebuttal. Their manuscript has considerably improved so I keep with the initial rating.

**Justification Of The Preliminary Rating:**

The work presented by Lalit M. and colleagues has shown to improve significantly the results obtained for instance segmentation. Mathematically speaking, it is a smart way of defining the instance problem. While the main idea is not novel, the authors have made an important work to adapt it to microscopy image processing both in 2D and 3D, and to make the method usable by the community. Not only they describe the method, but they also test it extensively with different datasets. The code released is 100% usable and transferable to other image datasets.

**Paper Type:**

both

**Special Issue:**

yes

---

> ### Author Response · Authors · 2021-03-17
> **Response to AnonReviewer3**
>
> We thank you for your very thorough reading  of our submission.
> We also thank you for your comments and questions.
> We address your suggestions one by one below:
>
> *  *"The authors should clean the manuscript and elaborate more on the methodology for its completeness. While most details remain similar to the ones defined in Neven et al., 2017, it is easier to follow the entire description when the parameters and variables are properly described."*
>
>     + We agree and have consequently made modifications to our main text which are uploaded and highlighted. Additionally we have modified *Appendix A*.
>
> * *"For example, the authors provide visualizations of the seediness scores and embeddings in their GitHub repository. Such visualizations could help to the understanding of their approach and would make clear what is their contribution with respect to the existing work. This information could be included in the Appendix."*
>
>     + We completely agree with this and have added some of these visualizations in *Appendix B*. Additionally we explicitly point to these resources which are indeed publicly available on GitHub.
>
> * *"`'The proposed objective encourages pairs of tags to have similar values iff the corresponding pixels belonged to the same object' Does this iff refer to if and only if? If so, please make a distinction. If not, please correct it."*
>
>     + Yes, *iff* refers to *if and only if*. We have replaced *iff* with *if and only if* in section *Related Work and Proposed Method* on page *2*.
>
> * *"'Greedily pick from all (remaining) pixels the one with the highest predicted seediness score $s_i > s_{min}$' what does remaining mean?"*
>
>     + Here, *remaining* refers to foreground pixels which have not yet been clustered, i.e. assigned to any instances. We have clarified this in the manuscript.
>
> * *"It probably makes no difference as it is a matter of scaling inside $S_k$, but strictly speaking, $x_{medoid}$ should include the 1/|$S_k$| factor to measure the average distance."*
>
>     + We agree with this - we have updated the formula on page *4* accordingly.
>
> * *"What is 3-Class referring to in the 3-Class UNet?"*
>
>     + Here, the 3 classes refer to Foreground, Background, and Membrane (border between foreground and background).
>     We have updated the text on page *6* to spell this out.
>
> * *"Table 1 and 2: Did you retrain all the methods except for Cellpose (public)? Please, order all the methods following the same order."*
>
>     + We did not train *Cellpose (public)* as you stated.
>     In addition, we did not train *PatchPerPix* and *Harmonic Embeddings* and have instead reported the numbers given in the original *PatchPerPix* and *Harmonic Embeddings* publication.
>     So far, we had sorted the methods based on their performance on the $AP_{0.5}$ metric. Seeing your comment we started wondering if that was a good choice and have now changed this such that an alphabetical order is always maintained with *Neven et al.* and *EmbedSeg* as the last two rows.
>
> * *"Why didn't you manage to calculate the GPU footprint of Neven et al. method?"*
>
>     + Thanks for pointing this out, this was a not intended mistake and is now fixed.
>
> * *"The description of Data and Data Handling in 3D could be improved with minimal effort: when providing the size of the images (e.g. 70 x 378 x 401) I assume the authors are talking about voxels, please indicate it as it was done with 2D data referring to pixels. 'we test on the right half of the second image', please indicate the voxel positions as done later with the Platynereis-ISH-Nuclei-CBG dataset."*
>
>     + These suggestions make a lot of sense. We changed the text according to your great suggestions.
>
> * *"In the training details, it is said: "All results obtained with EmbedSeg and the method by Neven et al. use the branched ERF-Net, ... (see Appendix A.4)", but then, in A.4, the shown architecture seems to be for the 3D datasets. Could you please explain this more clearly?"*
>
>     + This was indeed confusing and is now changed. *Appendix E* (previously labeled as *Appendix A4*) only shows the 3D architecture for *EmbedSeg*.
>
> * *"In the appendix, what are $\sigma_{k, 1}$, $\sigma_{k, 2}$ and $\sigma_{k, 3}$? what is $\sigma$?"*
>
>     + $\sigma_{k, 1}$, $\sigma_{k, 2}$ and $\sigma_{k, 3}$ are the predicted standard deviations for the object with label k along the x, y and z dimensions.
>     We have updated the text to reflect this on page *11* and Appendix *A*.
>
> * *"The final loss function (Eq. 6), seems to be a weighted sum. Could you please define $w_{seed}$, $w_{IoU}$ and $w_{var}$? Which values would you recommend? Did you use the same values in all the experiments?"*
>
>     + Yes, the final loss function is indeed a weighted sum.
>     We used the same values ($w_{seed}$ = 1, $w_{IoU}$ =1 and $w_{var}$ = 10) in all 2D and 3D experiments and included these values now in the manuscript.

---

> > ### Comment · AnonReviewer3 · 2021-03-19
> > **Answer to authors comments**
> >
> > I thank the authors for providing detailed answers and highlighting the changes in the new version of the manuscript. The manuscript has significantly improved.
> >
> > As it's been said, instance segmentation is extremely useful in microscopy image analysis. The approach chosen by the authors can provide competitive results and thanks to their great job making their code accessible and (re-)usable, I think many researchers could integrate it into their image processing workflows. Therefore, as a recommendation for this to happen, the authors could provide some intuition or hints about the selection of some parameters and how these are related to the characteristics of the microscopy data:
> > - The loss function uses different weights for the smoothing factor (w_var), and the balance between foreground&background(w_fg, w_bg). Is there any specific recommendation, such as it is done in the 3D case, for these choices?
> > - Do the authors recommend any maximum and minimum object size?
> >
> > This could be integrated either on their training notebooks or the GitHub repository.

---

> > > ### Author Response · Authors · 2021-03-19
> > > **Modifications on GitHub**
> > >
> > > We fully agree with the reviewer, our GitHub repo will evolve long after the paper is published and will, as many of our other methods, become much richer and comprehensive as we start having more users and include EmbedSeg in our teaching courses etc.
> > >
> > > Regarding the minimum object size, we have mentioned in our responses to reviewers that we now compute this value automatically from the given training data, hence, one hyperparameter gone! :)

---

### Official Review · AnonReviewer4 · 2021-03-09

**Confidence:** 3
**Preliminary Rating:** 3
**Final Rating:** 4

**Summary:**

The paper addressed the instance segmentation problem by modifying a previously state-of-the-art method Neven et al., 2019. The authors did two folds modifications, including changing centroid to medoid, and introducing the test-time augmentation. The experiments in both 2D and 3D datasets showed the effectiveness of the EmbedSeg method and the ablation studies are conducted accordingly.

**Strengths:**

1. The authors showed their insights into the instance segmentation problem in microscopy and improved marginally from the baseline method.
2. The authors released their codes and the results are reported on public datasets which helps to reproduce the results.
3. The experiments can be considered solid, and the authors are trying to compare many baseline methods in the discussion.

**Weaknesses:**

1. The improvements are marginal. Judging from the ablation study, it seems the test-time augmentation contributes much and the change of seed may sometimes introduce uncertainty.
2. The paper writing could be more clear, this may because the authors would want to include many discussions into the paper.

**Deanonymize Review:**

no

**Final Rating Justification:**

Thanks for the reply and I am glad to see this paper published in the coming MIDL.

**Justification Of The Preliminary Rating:**

The authors did show their insights into the task and the method they proposed is marginally better than the baseline network. However, the improvements are trivial and the structure of the paper is not easy to follow. I may rate the paper as weakly accept just to say that the paper can be improved from the present status.

**Paper Type:**

both

**Special Issue:**

no

---

> ### Author Response · Authors · 2021-03-17
> **Response to AnonReviewer4**
>
> We thank you for your comments on our approach.
> We also thank you for your suggestions. We address your suggestions one by one below:
>
> * *"The improvements are marginal. Judging from the ablation study, it seems the test-time augmentation contributes much and the change of seed may sometimes introduce uncertainty."*
>
>     + It is indeed true that on the datasets we use, *test-time augmentation* contributes more than the change of central embedding.
>     We strongly believe, though, that on more dense data, where object instances are tighter packed, and/or datasets where object shapes are more complex, the importance of the medoid embedding as we propose it will be increased.
>
>     Unfortunately, we did not find any available datasets that demonstrate this fact better than the ones we have used. Any recommendations would be very welcome.
>
> * *"The paper writing could be more clear, this may because the authors would want to include many discussions into the paper."*
>
>     + Thanks for this feedback. We have reworded multiple paragraphs in several sections of the manuscript in order to lend more clarity to the text.
>
>     Additionally, we believe that by addressing the remarks of other reviewers we have also improved the text with respect to your objections. Please let us know if this is not the case.
>
> * *"The structure of the paper is not easy to follow. I may rate the paper as weakly accept just to say that the paper can be improved from the present status."*
>
>     + Thank you for motivating us to improve the manuscript. As mentioned above, we believe to have at least partially succeeded and hope you agree that the manuscript is now better and more clear.

---

### Official Review · AnonReviewer1 · 2021-03-09

**Confidence:** 4
**Preliminary Rating:** 3
**Final Rating:** 4

**Summary:**

The proposed work extends Neven et al 2019 work on instance segmentation to 2D and 3D microscopic images. The proposed work has been thoroughly evaluated on standard publicly available datasets for both 2D and 3D images and shown to outperform the current state-of-the-art and baselines. The use of proposal free instance segmentation is an interesting idea presented in Neven et al 2019, which should be explored for biomedical image segmentation applications.

**Strengths:**

The extension of proposal-free instance segmentation of 2D and 3D images is an interesting idea worth exploring. Also, the paper contains a thorough experimental analysis with the current state-of-the-art methods, along with ablation studies.

**Weaknesses:**

In my opinion, the following points could further improve the quality of the presentation/work:

1) It would be better to summarize in Section 2, the modifications proposed in the current work as compared to the original work of Neven et al.  Similarly, it is expected to present the results of the original work (without any modifications) and the proposed work.

2) I am not very sure that presenting the results of semantic segmentation (UNet translated by connected components) would be of any help in comparing the performance of instance segmentation approaches. I would rather suggest comparing the performance with the more recent state of the art (if possible) "PolyTransform: Deep Polygon Transformer for Instance Segmentation"

**Deanonymize Review:**

no

**Final Rating Justification:**

The authors have incorporated the changes suggested and I am satisfied with the response

**Justification Of The Preliminary Rating:**

This application paper extends the interesting idea of proposal free instance segmentation to 2D and 3D microscopic images. The paper compares the proposed modified approach with the existing state of the art along with exhaustive ablation studies.  However, the presentation could be improved by clearly outlining the modifications proposed and comparing the results with the original work

**Paper Type:**

validation/application paper

**Special Issue:**

no

---

> ### Author Response · Authors · 2021-03-17
> **Response to  AnonReviewer1**
>
> We thank you for your comments on our approach. We also thank you for your suggestions. We address your suggestions one by one below:
>
> * *"It would be better to summarize in Section 2, the modifications proposed in the current work as compared to the original work of Neven et al. Similarly, it is expected to present the results of the original work (without any modifications) and the proposed work."*
>
>     + We changed the summary of our methodological contributions on page *7* in the section *Discussion*.
>
>     + Just for clarification, *Neven et al.* presented results on 2D, 8-bit RGB images containing city street scene objects belonging to multiple classes.
>     In the code presented on their project github webpage, some engineering changes had to be introduced for training on  gray-scale biological images; for dealing with 16-bit input raw images; for cases when instance segmentation masks were available as one-hot encoded (for example, with the *BBBC010-2012* dataset).
>     Also, the minimum object size considered by *Neven et al.* was set to a constant value of 128 pixels and the seediness threshold used for picking out instance centers was similarly set to a constant value of 0.5.
>     Furthermore, the authors did not provide any 3d network implementation or any 3d results using their method, as this was not needed for their application of interest.
>
>     Hence, we could only report numbers using the original method by *Neven et al.* only 2d data. Additionally, we were forced to make slight modifications in order to operate on the *BBBC010-2012* label data, which was one-hot encoded.
>     Furthermore, so as to not underplay performance of *Neven et al.*, we have ensured that both *Neven et al.* and *EmbedSeg* follow the same training regimen and the same choice of parameters during inference, in order to highlight the impact of our methodological contributions (see above).
>
> * *"I am not very sure that presenting the results of semantic segmentation (UNet translated by connected components) would be of any help in comparing the performance of instance segmentation approaches.
> I would rather suggest comparing the performance with the more recent state of the art (if possible) 'PolyTransform: Deep Polygon Transformer for Instance Segmentation'"*
>
>     + The reason why we chose to include the *3-Class U-Net* results is because the U-Net is very well known and remains a frequently used method in the medical imaging field.
>
>     Thank you for pointing us to *PolyTransform*, a method we did not encounter before.
>     We initially contemplated to include *PolyTransform* as an additional baseline, but quickly discovered that it requires some instance masks (obtained by any other method) as additional input.
>     Additionally we could not locate any available open source code for *PolyTransform* for us to try this method out.
>     While we think it would in principle be interesting to see how *PolyTransform* would improve results by *EmbedSeg*, *StarDist*, or any other of our baseline methods, we don't think such experiments are a good fit for the current story in our manuscript.
>
> * *"The presentation could be improved by clearly outlining the modifications proposed and comparing the results with the original work"*
>
>     + We summarized our methodological contributions on page *7* section *Discussion* in the manuscript and have now tried to give additional emphasis on this section.
>
>     Additionally, we believe that by addressing the remarks of other reviewers we have also improved the text with respect to your objections. Please let us know if this is not the case.

---

### Meta-Review · Area_Chair1 · 2021-03-29

**Recommendation:** Accept (Oral & Special Issue Candidate)

**Metareview:**

The reviewers are consistent with their ratings and comment that this work is interesting and well presented. Plus points are given for reproducible research through the author's release of source code and datasets. It should be considered for oral presentation.

**Paper Type:**

both

---

### Decision · Program_Chairs · 2021-03-31

**Decision:**

Accept

**Comment:**

Congratulations your paper has been selected as a long oral.